# Roles of Granular Sludge Size Restricting and Organic Degradation in an Extended Filamentous AGS System Using Agnail Aeration Device

**Jun Liu [1,2], Songbo Li [3], Weiqiang He [1], Qiulai He [4], Xiangzhou Meng [5], Liangliang Wei [6], Heng Liang [6,*] and Jun Li [7]**

1 School of Modern Agriculture, Jiaxing Vocational & Technical College, Jiaxing 314036, China; liuj521282@sina.com (J.L.); hwq2005003@163.com (W.H.)
2 Department of Civil Engineering, Tongji Zhejiang College, Jiaxing 314051, China
3 School of Modern Urban Construction, Jiaxing Vocational & Technical College, Jiaxing 314036, China; lly202204@sina.com
4 Department of Water Engineering and Science, College of Civil Engineering, Hunan University, Changsha 410082, China; qiulaihe@hnu.edu.cn
5 College of Environmental Science and Engineering, Tongji University, Shanghai 200092, China; xzmeng@tongji.edu.cn
6 State Key Laboratory of Urban Water Resource and Environment, Harbin Institute of Technology, Harbin 150090, China; weill333@163.com
7 College of Environment, Zhejiang University of Technology, Hangzhou 310014, China; tanweilijun@zjut.edu.cn
* Correspondence: hitliangheng@163.com

**Abstract:** This work investigated the role of an agnail device (manually made from a comb) on sludge size restriction and organic degradation in extended filamentous aerobic granular sludge-sequencing batch reactors (AGS-SBRs) with artificial wastewater. Two identical SBRs (R1 and R2) were employed in this experiment. Extended filamentous AGS with a large size was achieved in both SBRs by seeding the dewatering the sludge on day 40. R1 (the control) did not use the agnail aeration device, and the extended filamentous AGS system was finally disintegrated. However, R2 promptly employed the agnail device on days 56–59, the extended filamentous AGS size obviously decreased from 4.8 mm to 2.5 mm, and the dominant filamentous species, including *Proteobacteria*, *Acidobacteria*, and *Choroflexi*, gradually shrank at a low level, acting as a framework for AGS recovery. This was because enough nutrients diffused into the inside of small sludge for the filamentous living. Simultaneously, the sludge volume indexes ($SVI_5$ and $SVI_{30}$) sharply decreased from 155.8–103.9 to 51.7–46.6 mL/g, and the mixed liquid suspended solids (MLSSs) and extracellular polymeric substances (EPSs) in R2 both increased and were kept at 5816 mg/L and 69.1 mg/g·MLVSS, respectively. These contributed to enhancing the sludge's structural stability to avoid AGS failure. COD and $NH_4^+$-N in R2 were both degraded by simultaneous nitrification and denitrification (SND) processes throughout the experiment, which was not significantly influenced before or after the agnail aeration device was employed. These results indicate that the agnail device can effectively restrict AGS size and limit the extended filamentous overgrowth with nutrient diffusion into the sludge's interior, which can prevent AGS disintegration. In addition, this device had no significant influence on organic degradation.

**Keywords:** aerobic granular sludge; extended filaments; agnail aeration device; organic degradation

## 1. Introduction

As a novel biotechnology, aerobic granular sludge (AGS) has been widely researched in bioreactors for decades [1–4]. This is because of its advantages of compact structure, excellent settleability, resource recovery, and low energy usage [5–7]. AGS structural

stability still presents a huge challenge for long-term operation and organic degradation. Sludge size plays an important role in AGS stability [8–10].

Presently, AGS has been successfully obtained in sizes ranging from 0.02 to 9.0 mm [11,12]. Sludge size significantly affects the characteristics of extracellular polymeric substances (EPSs), oxygen consumption, and simultaneous nitrification and denitrification (SND) [13–15]. Moreover, numerous studies have demonstrated that AGS with a large size tends to hinder mass transfer limitation, diffusion, and nutrient utilization, which decreases the metabolic activity of functional microbes, causes cavity structures, and weakens AGS structural stability; this subsequently leads to granular sludge disintegration under hydrodynamic shear stress [16–18]. Thus, most researchers have proposed that the best particle size range is 0.2–3 mm by studying AGS properties (settling performance and steady-state operation) with different particle sizes in the bioreactors [19]. Furthermore, the excess growth of filamentous microbes is another chief factor for AGS disintegration [20,21]. It has been reported that pH, dissolved oxygen (DO), and nutrients are related to filamentous growth [22,23]. Thus, the relationship between AGS with a large size and filamentous overgrowth is relevant; however, it is rarely reported.

Recently, studies on restricting AGS size or filamentous excess growth have been conducted. Strategies for sludge size restriction have been reported, including improving hydraulic shear force [24,25], removing the large size with different methods [9,26], and using a spiny aeration device [27]. In addition, some methods of filamentous overgrowth control have been employed, including NaClO, $Cl_2$, or $H_2O_2$ addition; sludge micropowder, particulate substrates, ballasting agents. and coagulant addition [20,28–30]. Improving the shear force, organic loading rate, and salinity have also been attempted to mitigate filamentous bacterial growth [31–34]. Although these works have been archived, they have solely focused on AGS size or filamentous overgrowth. Effectively controlling sludge size is very important for the stable running of AGS in an extended filamentous granular sludge system, but similar work has rarely been reported at present.

In brief, the primary objective of this work was to investigate the influences on sludge size restriction and organic degradation in the filamentous AGS system using the agnail aeration device. The AGS characteristics of biomass, size, settling velocity, microbial composition, and EPS were also carried out throughout the experiment.

## 2. Materials and Methods

### 2.1. Modification of Agnail Aeration Device and Batch Experiment

Combs made of plastic (Figure 1a) are usually used for one's hair, and long hair falls during the dressing process. Then, it was remolded by manual work (Figure 1b), including the modification of both ends with two holes of 9 mm. Finally, the agnail device was successfully obtained (Figure 1c). This device easily traversed the air pipe (D = 8 mm) and was then put into the bioreactor (H × D = 100 cm × 10 cm) from the top in this work.

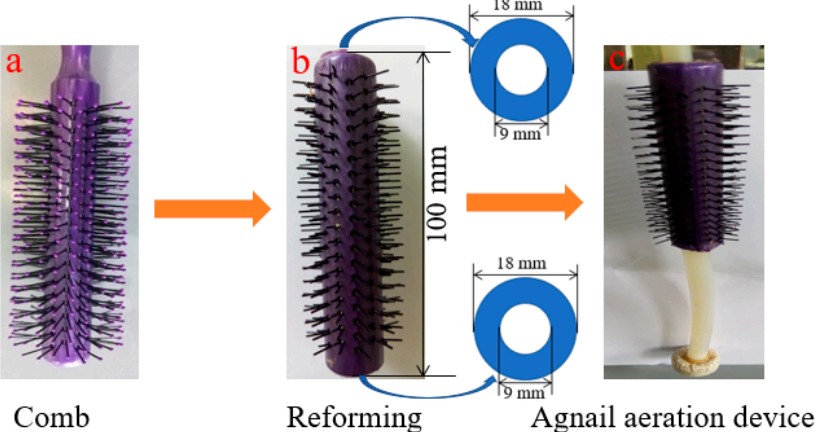

**Figure 1.** Schematic diagram of agnail aeration device by manufacture modification.

The aim of the batch experiment was to test the efficiency of the agnail aeration device (Figure 1c) for restricting large AGS and the organic removal rate. AGS with a large size (Figure 2a) was from one lab-scale SBR and used for a 4-day batch experiment. The influent was artificial wastewater, and an SBR made of plexiglass with an effective volume of 4 L (H × D = 100 cm × 10 cm) was used. The wastewater characteristics and SBR operation were presented in 2.2 SBR operation and 2.3 wastewater.

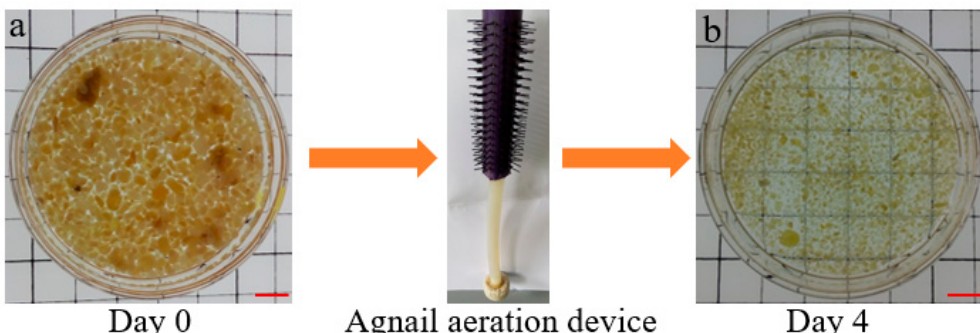

**Figure 2.** Pictures of AGS with large size in the batch experiment using agnail device, scale bar: 1 cm.

## 2.2. SBR Operation

Two SBRs (R1 and R2) made of plexiglass with the same working volumes of 4 L (H × D = 100 cm × 10 cm) were used in this work. R1, the control adopted the conventional aeration device; R2 adopted the agnail aeration device (Figure 1c) on days 56–59, and at other times it was not used. Two fine air pumps (PL60, RESUN, China) were used to supply aeration to the bottom of the SBRs through air diffusers with a uniform airflow rate of $0.1\ m^3 \cdot h^{-1}$ (superficial gas velocity = $0.44\ cm \cdot s^{-1}$) and a 50% volumetric exchange ratio. Each 160 min cycle consisted of the following steps: feeding for 5 min, aeration for 120 min, settling for 2 min, discharge for 5 min, and idle time for 28 min. All pumps and valves were controlled by a PLC (programmable logic controller) (Hangzhou Zhijiang Water Processing Equipment Co., Ltd., Hangzhou, China) in the experiment.

## 2.3. Wastewater and Seeding Sludge

Both reactors (R1 and R2) were fed with acetate-containing synthetic wastewater. The artificial wastewater had an influent chemical oxygen demand (COD), nitrogen from ammonium ($NH_4^+$-N), and total phosphorus (TP) maintained at 500–700 $mg \cdot L^{-1}$, 25–35 mg/L, and 10–15 $mg \cdot L^{-1}$, respectively. The wastewater was manually made once every three days, and it was stored in a sealed tank with a pump for stirring. A trace element solution was also added to the wastewater. The composition of the synthetic wastewater and its enrichment in trace elements can be found in a previous paper [35]. The dewatering sludge from Qige WWTP (Hangzhou, China) was screened using a sieve with a size of 0.15 mm. A quantity of 0.2 L of dewatering sludge (≤0.15 mm) was inoculated into each SBR. The indoor temperature was in the range of 26 ± 4 °C using air-conditioning (Gree Group Co., Ltd., Zhuhai, China) throughout the experiment.

## 2.4. Analytical Methods

The chemical oxygen demand (COD), ammonium ($NH_4^+$-N), nitrate ($NO_3^-$-N), nitrite ($NO_2^-$-N) mixed liquor (volatile) suspended solid (ML(V)SS), and SVI ($SVI_5$ and $SVI_{30}$) values were measured using standard methods [36]. EPSs were extracted using the formaldehyde–NaOH method [37]. The polysaccharide (PS) concentration was determined by the phenol-sulfuric acid method using glucose as the standard [38]. The protein (PN) concentration was determined by Coomassie brilliant blue using bovine serum albumin (BSA) as the standard [39].

Six sludge samples (S1–S6) were regularly collected on days 0, 30, 55, 59, 70, and 90. Each sample was randomly selected three times and then mixed into one tube (10 mL).

All tubes were stored in a refrigerator at $-20\ ^\circ$C for microbial analysis. DNA extraction was performed using the E.Z.N.A™ Mag-Bind Soil DNA Kit (OMEGA, Los Angeles, CA, USA) according to the manufacturer's instructions. Microbial analysis was conducted at Sangon Biotech (Shanghai Co., Ltd., Shanghai, China) using an Illumina Miseq 2 × 300 (PE300, San Diego, CA, USA). About 10 ng of DNA from the sludge sample was used for 16S rRNA sequencing targeting the hypervariable V3–V4 regions. The data were processed by the Quantitative Insights into Microbial Ecology (Qiime) pipeline (http://qiime.org/, accessed on 18 October 2016) and Mothur (http://www.mothur.org/, accessed on 18 October 2016) in this work. Bacterial analysis was carried out using high-quality sequences with an identity of more than 97% of the threshold. The details were reported in previous work [40].

## 3. Results and Discussion

### 3.1. Effects of Batch Experiment

Figures 2 and 3 show the results of the agnail aeration device (Figure 1c) for controlling large AGS and the organic removal rate. AGS with a large size (mean size = 3.5 mm, Figure 2a) became a small sludge (mean size = 1.5 mm, Figure 2b) in the SBR with the agnail aeration device on day 4. The AGS size distributions had outstanding changes; the proportion of AGS sizes of 4–5 mm, 3–4 mm, and 2–3 mm was 15 ± 0.5%, 25 ± 0.5%, and 55 ± 0.5%, which decreased to 2 ± 0.5%, 5 ± 0.5%, and 41 ± 0.5%, respectively (Figure 3a). However, the proportions of AGS size of 1–2 mm, 0.2–1 mm, and 2–3 mm increased from 10 ± 0.5% and 0% to 44 ± 0.5% and 8 ± 0.5% after a 4-day batch experiment (Figure 3b). The pollutants' variations in COD and $NH_4^+$-N were also assessed; the influent COD was 566–599 mg/L, and the effluent COD increased from 0 to 53.2 mg/L (Figure 3c). However, the influent and effluent $NH_4^+$-N stayed at 25.9–27.1 mg/L and 0 mg/L (Figure 3d). These results indicate that the agnail aeration device can effectively control AGS size, and it had a slight influence on COD removal besides $NH_4^+$-N. This is consistent with previous work [27].

### 3.2. Variations of AGS Properties

As shown in Figure 4, the dewatering sludge with a size below 1.5 mm was used as the seeding sludge in R1 and R2; the sludge presented a black-brown color and was regulated sharply (Figure 4a,g). After 10 days of running, an AGS with a brown color and elliptic sketch was rapidly formed and became predominant in both SBRs (Figure 4b,h). This suggested that the AGS was quickly cultivated by seeding the dewatering sludge, which agrees well with earlier studies [41,42]. Along with microbial growth, AGS with a large size was successfully obtained, and the extended filamentous bacteria were observed in the AGS system on day 40 (Figure 4c,i). After 15 days of operation, the filamentous bacteria of the large AGS grew on the sludge surface. This was due to the large AGS limiting the substance diffusion, which led the filamentous bacteria to extend to the AGS surface to live [16,17,23,24]. The agnail aeration device (Figure 1c) was employed in R2 on days 56–59, and this device was not used in R1 for the whole experiment. After that, the extended filamentous bacteria grew excessively and became the dominant in R1 on day 60 (Figure 4e); the large AGS began to break on day 70 (Figure 4e) and thoroughly disintegrated on day 90 (Figure 4f), which indicated that R1 failed in this experiment. However, the large AGS in R2 obviously became small granules on day 60 (Figure 4j), the extended filamentous bacteria did not grow excessively when enough nutriments were diffused into the sludge interior, the AGS slowly recovered from day 70 to day 90 (Figure 4h,l), and R2 was finally stable in the end. This indicated that the agnail aeration device restricted the AGS size and prevented the sludge from disintegrating from the filamentous overgrowth in this work.

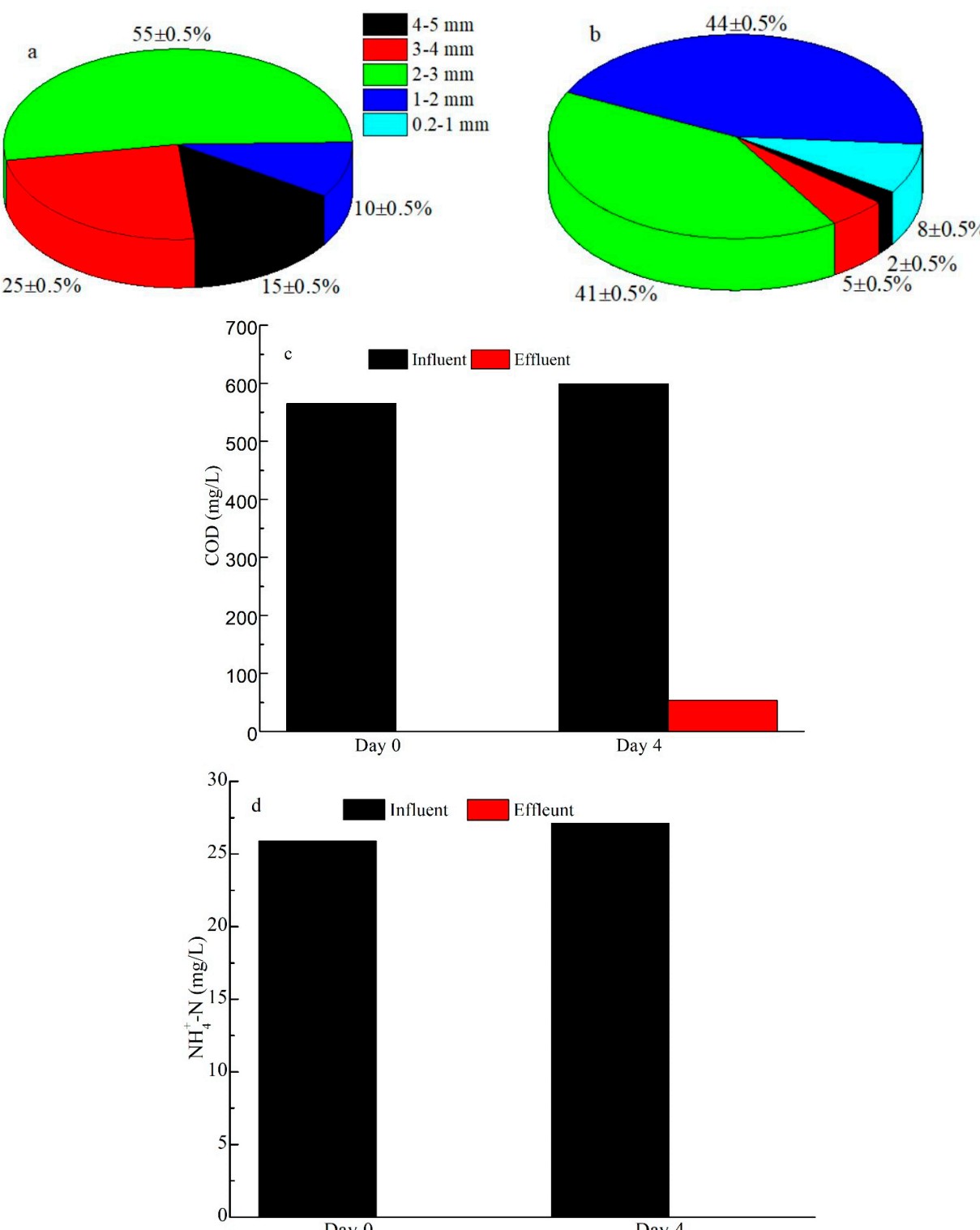

**Figure 3.** Effects of the batch experiment on AGS size distribution before (**a**) and after restriction (**b**); COD (**c**), and NH-N (**d**) removal rates using agnail aeration device.

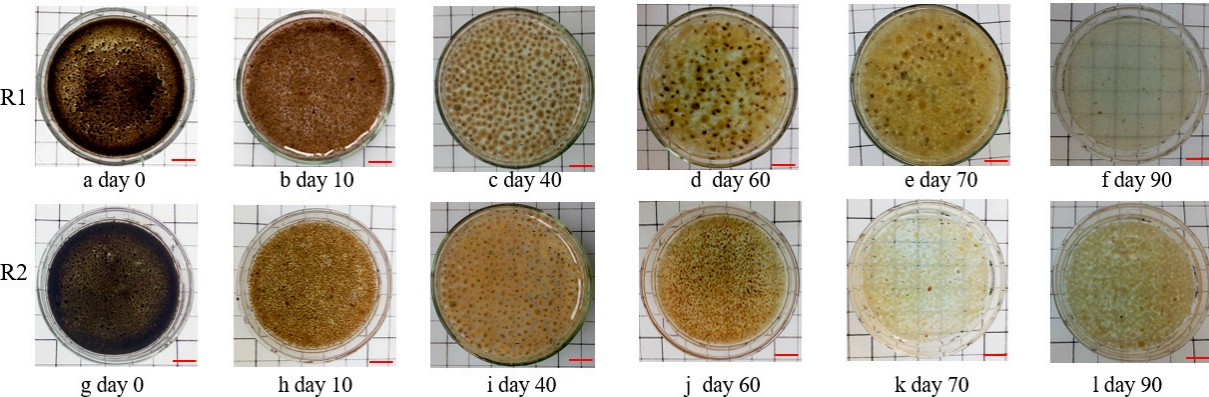

**Figure 4.** Images of sludge in R1 and R2 in the whole experiment; scale bar, 1 cm.

The initial MLSS and MLVSS of R1 and R2 were 3543–3573 mg/L and 1613–1633 mg/L, suggesting that the seeding sludge (dewatering sludge) contained about 55% inorganic matter. Then, these values in both SBRs showed an increasing trend, while $SVI_5$ and $SVI_{30}$ had no significant change before day 40 (Figure 5). In addition, the AGS in R1 and R2 were both clearly observed with a regulated sketch after 10 days of operation, and the extended filamentous bacteria appeared on the AGS surface on day 40 along with an increase in sludge size (Figure 4). Subsequently, some biomass began to be washed out from the bioreactors; MLSS and MLVSS decreased, while SVI increased. In R1, $SVI_5$ and $SVI_{30}$ quickly increased from 51.9–51.9 mL/g (day 40) to 226.8–175.5 mL/g (day 90) with excessive filamentous bacteria growth (Figure 4), indicating that the sludge's settling ability became worse. As a result, the MLSS and MLVSS values in R1 both decreased from 6762–6251 mg/L (day 40) to 616–554 mg/L (day 90). The agnail aeration device was employed in R2 on days 56–59; the AGS size was effectively restricted, more small granules were presented on day 60 (Figure 4j), and the filamentous bacteria did not excessively proliferate as nutriments (C, N, P, DO) diffused into the small sludge interior. Meanwhile, $SVI_5$ and $SV_{30}$ values began to decrease from 155.8–103.9 mL/g (day 55) to 51.6–46.6 mL/g (day 90), and MLSS and MLVSS began to increase from 4036-3462 mg/L (day 55) to 5816–5293 mg/L (day 90) (Figure 5). This means that the gap between $SVI_5$ and $SVI_{30}$ was reduced in the extended filamentous bacterial AGS with a large size using the agnail aeration device in this work. These experimental results demonstrate that this agnail device could successfully restrict large AGS [27] and further prevent filamentous overgrowth for long-term operation in this work.

The sludge size and settling velocity in R1 and R2 had similar variation trends before 50 days; the two values had obvious changes from day 56 to day 90 (Figure 6). In R1, the extended filamentous bacteria grew excessively, with the sludge size increasing from 1.5 mm to 4.8 mm (the maximum on day 60). Subsequently, the AGS settling velocity of R1 had a downtrend, as sludge breakage occurred with no timely control of sludge size throughout the whole experiment (Figure 6). As the agnail device was promptly employed in R2 on days 56–59, the sludge size rapidly decreased from 4.8 mm (day 50) to 2.5 mm (day 60), then slowly increased and stayed at 3.2 mm on day 90. Meanwhile, the sludge settling velocity had a similar tendency on days 60–90 in R2 (Figure 6). As a metabolic production source, EPS plays a crucial role in AGS formation and structure [5,43]. In this view, the concentrations of PN and PS in EPS were measured throughout the experiment (Figure 7). Briefly, the EPS contents in the two SBRs both decreased from 111.1 mg/(g·VSS) (the seeding sludge, day 0) to 70.9 mg/(g·VSS) (day 20). This was due to the dewatering process making surplus sludge containing high levels of EPS and the microorganisms in the dewatering sludge with artificial wastewater making sludge containing low levels of EPS. With AGS development, the EPS contents in both SBRs reached 77.0 mg/(g·VSS) (day 30); after that, the filamentous bacteria extended on the AGS surface as the sludge size increased on day 40 (Figure 4c,i). The EPS showed a decreasing

trend with sludge disintegration on days 50–90 in R1; meanwhile, PN and PS both decreased from 27.2–40.1 to 11.8–27.1 mg/(g·VSS) (day 50–90), and the PN/PS ratio was unbalanced (Figures 4d,f and 7). However, the sludge size was effectively restricted by the agnail device on days 56–59 in R2, and the AGS gradually recovered, with the extended filamentous bacteria fading away on days 60–90 (Figures 4j–l and 7). Then, the EPS, including PN and PS, had slow growth, with AGS recovering on days 60–90 in R2; meanwhile, these values stayed at 69.1, 31.8, and 37.3 mg/(g·VSS), respectively, on day 90. This means that the PN/PS ratio had a smaller gap in R2 than in R1. As we know, PN with a 3D structure plays an important role in sludge skeleton structure, and PS as a biogel with stickiness plays a key role in bacterial adherence [5,43]. Therefore, the PN/PS ratio stayed within a suitable range for AGS formation and stability [8]. From this view, the agnail device effectively restricted the sludge's size, prevented filamentous overgrowth, and maintained the PN/PS balance for long-term AGS operation in this work.

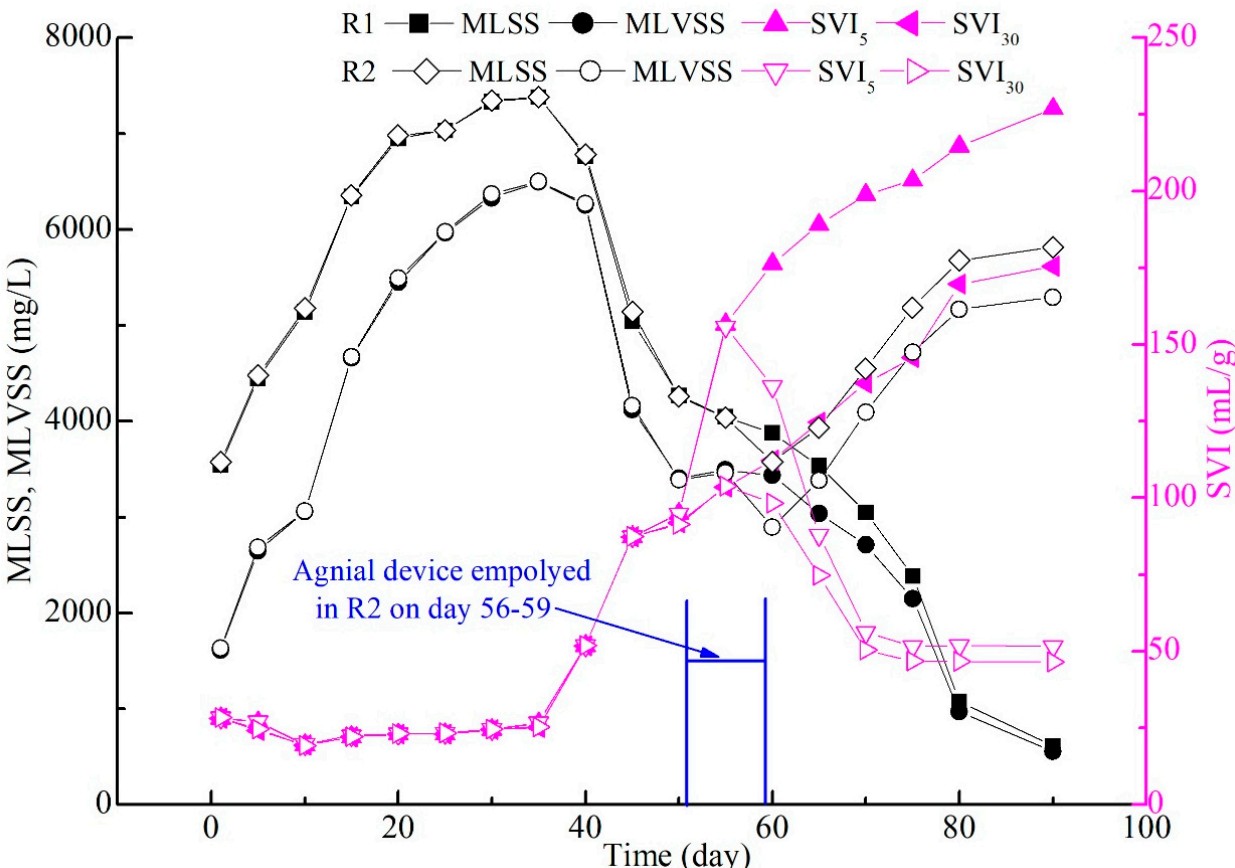

**Figure 5.** Changes in MLSS and SVI in the whole experiment.

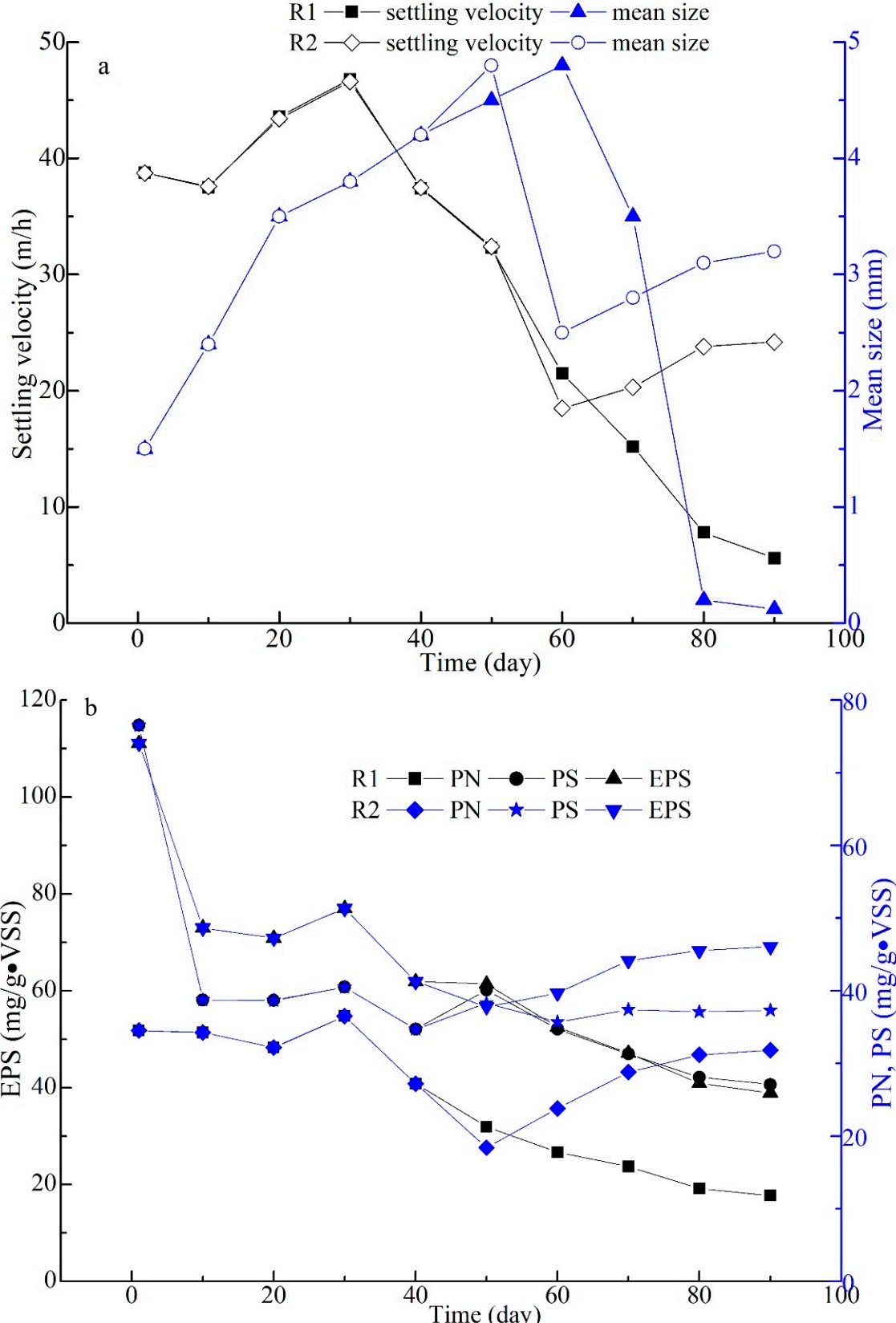

**Figure 6.** Changes of sludge characteristics on velocity & size (**a**), EPS (**b**) in the whole experiment.

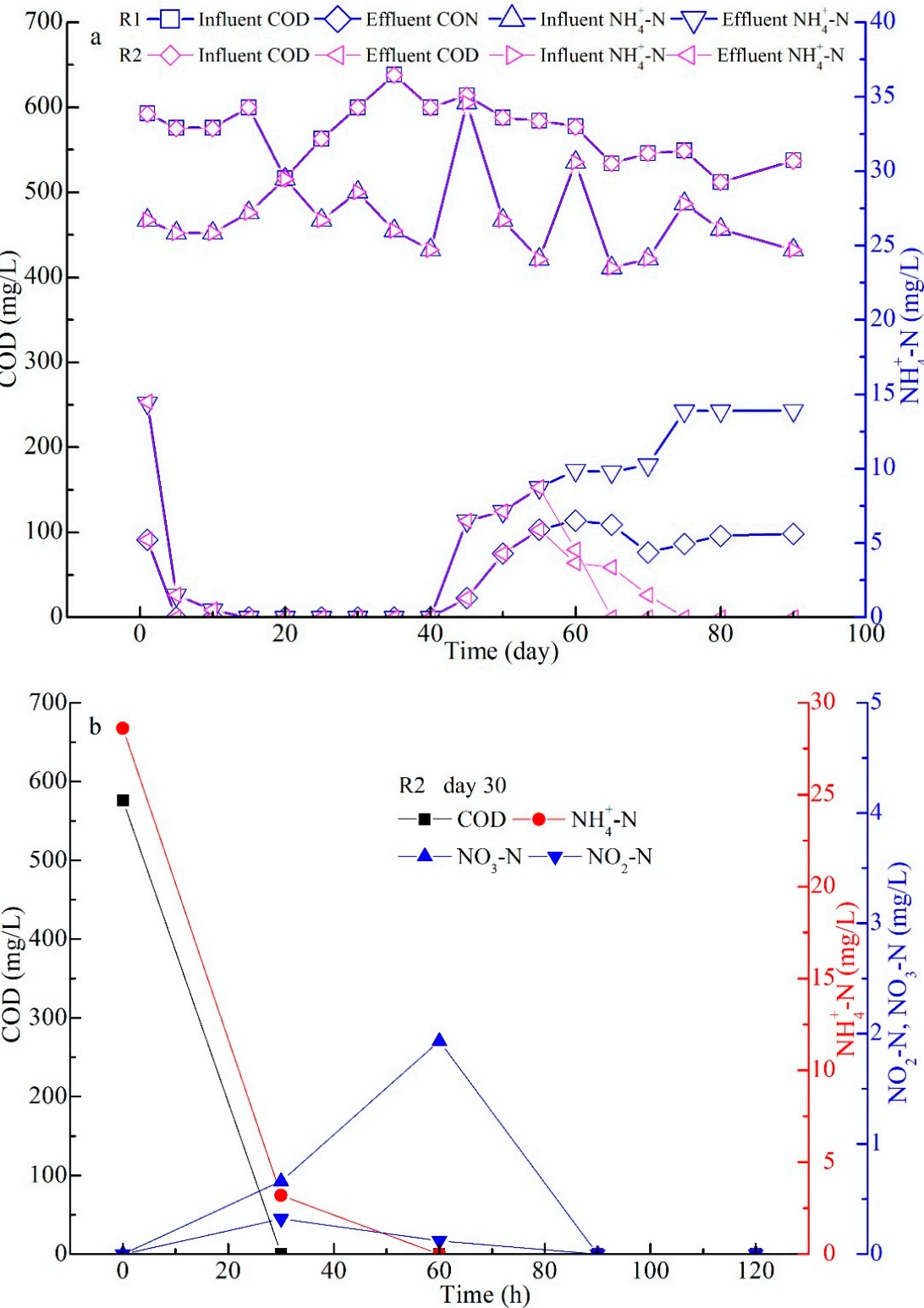

**Figure 7.** *Cont.*

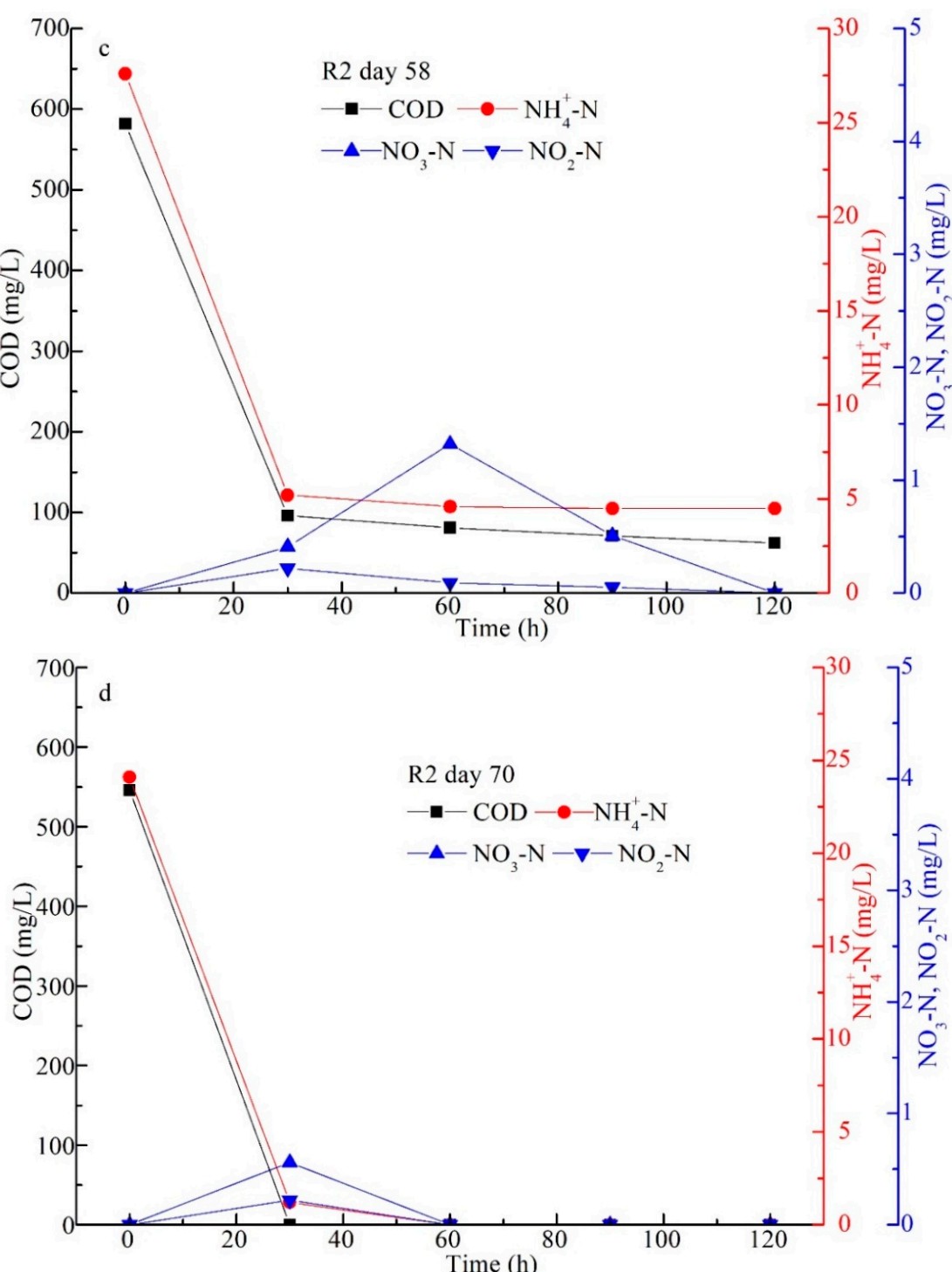

**Figure 7.** Performance in the whole experiment (**a**); organic degradation in the typical cycles on day 30 (**b**), day 58 (**c**), and day 70 (**d**), respectively.

### 3.3. Performances of the Bioreactors

The performances of the COD and $NH_4^+$-N removal rates are shown in Figure 7a. Throughout the experiment, the influent COD and $NH_4^+$-N started at concentrations of 500–700 mg/L and 25–35 mg/L, respectively. After the seeding process, the effluent COD and $NH_4^+$-N both reached 0 mg/L on day 5, indicating that the biomass in the seeding sludge was quickly reactive in a suitable environment. In addition, the two values stayed at 0 on day 40, and then they both showed an increasing trend on day 45 when

the extended filamentous bacteria grew excessively with the sludge size increasing in this work. Consequently, the AGS settling property deteriorated, and some sludge was washed out from the two SBRs on days 45–55 (Figures 4 and 5). Because effective action occurred in R1, the filamentous bacteria quickly developed, and the AGS disintegrated (Figure 4d–f). Finally, the effluent COD and $NH_4^+$-N both showed an increasing trend, and the two values arrived at 109 and 13.9 mg/L on day 90 (Figure 7a). However, the effluent COD and $NH_4^+$-N both decreased on days 60–90 after the agnail device (Figure 1c) was employed in R2 on days 56–59. These results suggest that oversize AGS induces the filamentous bacteria to grow excessively and leads to sludge disintegration, which in turn negatively affects organic degradation [22,30]. This also proves that the agnail device effectively controlled the extended filamentous AGS size and maintained system performance in organic degradation in this work.

Figure 7b–d show the variations in COD, $NH_4^+$-N, $NO_3^-$-N, and $NO_2^-$-N in R2 in one cycle on days 30, 58, and 70, respectively. The results suggested that COD and $NH_4^+$-N were almost entirely eliminated in 30–60 min of operation; then, $NO_3^-$-N and $NO_2^-$-N both increased from 0 to 1.93 mg/L (60 min) and 0.32 mg/L (30 min) and then decreased to 0 mg/L (90 min) on day 30, presumably due to AGS system running stably and R2 having a high amount of MLSS in R2 (Figure 5). This indicated that simultaneous nitrification and denitrification (SND) were achieved in R2 due to the large AGS with a mean size of 3.8 mm (Figure 6a). On day 58, SND was also obtained, but the effluent COD and $NH_4^+$-N were 60 and 4.5 mg/L (Figure 7c), which were higher than those on day 30 (Figure 7b). This was related to the coexistence of extended filamentous bacteria in R2 even though the agnail device (Figure 1c) was employed; then, some of the AGS was washed out at a relatively lower MLSS than that on day 30 (Figure 5). On day 70, the effluent COD and $NH_4^+$-N were almost entirely eliminated in 60 min of operation; then, $NO_3^-$-N and $NO_2^-$-N both increased from 0 to 0.56 mg/L (60 min) and 0.22 mg/L (30 min) and then decreased to 0 mg/L (60 min) (Figure 7d). This means that the SND process was also successfully obtained after AGS was oversized, and the extended filamentous bacteria were effectively controlled in this experiment. The experimental results demonstrate that the agnail device had no obvious effect on organic degradation through the SND process. This is agreed well with our batch experiment and a previous study [27].

*3.4. Microbial Community Analysis*

In this work, the microbial communities from six sludge samples (S1–S6) from R2 were analyzed through high-throughput sequencing at the phylum level. As shown in Figure 8, the most predominant phylum in the six sludge samples was *Proteobacteria*, with relative abundances of 33.7% (S1), 67.3% (S2), 72.9% (S3), 54.4% (S4), 50.1% (S5), and 50.9% (S6), respectively. This was found to be the most abundant phylum in AGS and wastewater treatment plants (WWTPs) with functions of EPS production and organic removal [40,44]. In addition, the percentage of *Proteobacteria* in S1–S6 first increased (S1–S3) and then tended to decrease (S4–S6), indicating that most *Proteobacteria* belonged to the extended filamentous bacteria [21,22], and they were tentatively controlled by the agnail device on days 56–59. With the AGS size reduction, the nutrients could diffuse into the AGS interior, and the filamentous bacteria grew to balance it (S5 and S6), acting as a framework for AGS reformation; finally, the normal AGS recovered with sharp integration (Figure 4l). The other main phyla groups for S1–S6 were *Firmicutes*, *Bacteroidetes*, *Acidobacteria*, and *Choroflexi* with different abundances, which also had similar variation trends in this work (Figure 8). Generally, *Proteobacteria*, *Acidobacteria*, and *Choroflexi,* the typical filamentous bacteria, have been frequently found in sludge bulking processes [21–23], summing to 72.9%, 8.2%, and 7.0%, respectively, in extended filamentous AGS with a large size (S3). After the agnail device was employed, these phyla dropped and reached a balance in the sludge samples of S4–S5 with small sizes. These results indicate that the filamentous bacteria of *Proteobacteria*, *Acidobacteria,* and *Choroflexi* are the major contributors to extended filamentous overgrowth for large AGS. *Firmicutes* of the seeding sludge (S1) possessed

the highest relative abundance of 42.8%, but it quickly decreased to 8.1% (S3) and then increased and stayed at 20.8% (S6) (Figure 8). This was related to the change in wastewater and the agnail device employed. In addition, *Firmicutes* belongs to the facultative anaerobes, and AGS recovered with a regular structure to provide a suitable microenvironment, which contributed to microbial growth with a high abundance in this work [20,22,23]. Therefore, the agnail device effectively restricted AGS size and prevented filamentous overgrowth to maintain sustainable operation in this experiment.

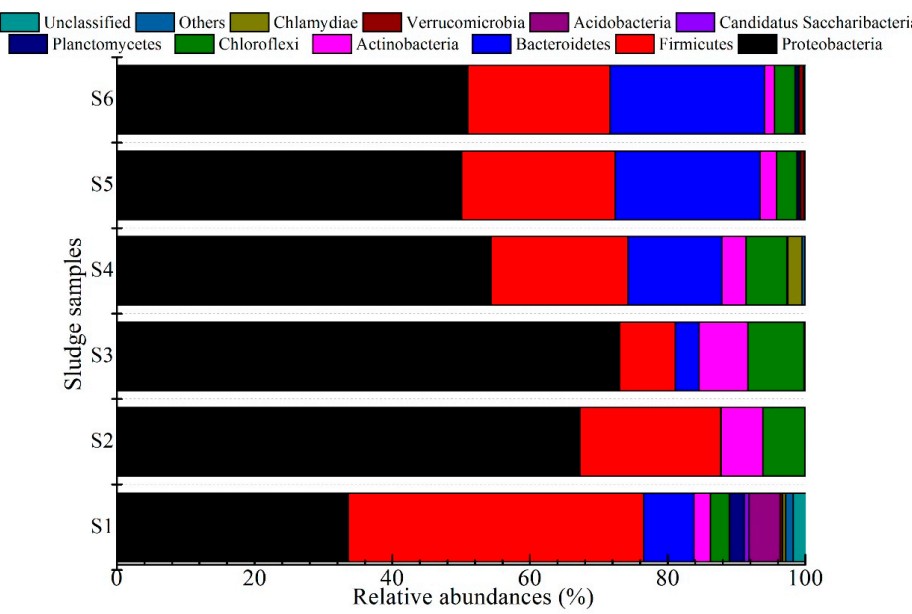

**Figure 8.** Microbial compositions in all sludge samples at the phylum level.

### 3.5. Possible Roles of Agnail Device on Large AGS System

In this work, AGS with a large size was rapidly formed by seeding the dewatering sludge with enough aggregates in the start-up stage (Figure 4a,b,g,h) [42,43]. After microbial adaptation with abundant nutriments, they quickly grew with easily degradable substances (acetate, $NH_4Cl$, $KH_2PO_4$), especially the filamentous bacteria [12,18]. With increasing sludge size, nutrient diffusion was limited [13,14], and the filamentous bacteria inside the sludge penetrated outside the AGS for foraging; subsequently, the extended filamentous AGS with a large size was clearly observed in this experiment (Figure 4c,i) [18,20,22]. Then, the filamentous bacteria of *Proteobacteria*, *Acidobacteria*, and *Choroflexi* with a high abundance became the dominant phylum in the AGS (Figure 8). After the timely employment of the agnail device in the bioreactor, the AGS size was effectively restricted (Figures 4 and 6a); some filamentous bacteria were exposed to rich nutrition, did not grow excessively, and were maintained at a low level (Figure 8, S4–S6); another small sludge reaggregated with the filamentous bacteria as the framework; and EPS (PN/PS) showed increasing trends and stabilized with a balanced ratio of PN/PS (Figure 6b), which contributed to maintaining the AGS structure for rapid recovery [5,44]. Concurrently, the organic degradation of COD and $NH_4^+$-N was successful through the SND process before and after employing the agnail device (Figure 7b–d); the SND efficiency was slightly influenced by the filamentous bacteria existing (Figure 7c), and it recovered after the biomass gradually increased with AGS system recovery (Figure 7d). This indicated that employing the agnail device had no significant negative role in organic degradation [27].

### 4. Conclusions

An agnail aeration device manually remolded from a comb was employed in the extended filamentous AGS system. The size of the extended filamentous AGS was effectively controlled by the agnail device, and the extended filamentous bacteria did not grow

excessively due to nutriment diffusion into the small granular sludge. Meanwhile, the microbial analysis indicated that the predominant filamentous abundance of *Proteobacteria*, *Acidobacteria*, and *Choroflexi* sharply reduced at a low level, which in turn acted as a framework for AGS recovery with adequate EPS. Meanwhile, organic degradation was successfully obtained by the SND process in this work, which was not distinctly affected by employing the agnail device. Briefly, the agnail device effectively controlled AGS size and prevented filamentous overgrowth for steady operation. In addition, this device had no significant influence on organic degradation.

**Author Contributions:** Data curation, validation, and writing—original draft preparation, J.L. (Jun Liu), S.L. and W.H.; review and editing, Q.H. and L.W.; funding acquisition, resources, supervision, review and editing, J.L. (Jun Liu), H.L., X.M. and J.L. (Jun Li). All authors have read and agreed to the published version of the manuscript.

**Funding:** The authors would like to acknowledge the financial support from the Open Project of the State Key Laboratory of Urban Water Resources and Environment, Harbin Institute of Technology (HC202156), Science and Technology Planning Project of Jiaxing (2021AY10080, 2021AD30166).

**Data Availability Statement:** Not applicable.

**Conflicts of Interest:** The authors declare no conflict of interest. This study does not include experiments conducted on humans or animals.

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
