# Peer review of "Roles of Granular Sludge Size Restricting and Organic Degradation in an Extended Filamentous AGS System Using Agnail Aeration Device"

_water, doi:10.3390/w15112009_

Round 1

Reviewer 1 Report

The article is valuable. However, the Authors make mistakes that should be corrected. 

Figure 3 a - unclear description

Figure 5 - vertical axis SVI (g/mL) ?

Line 256 Figure 8 b - d ?

Line 275 - 277 Figure 7 ?

Line 286 (Figure 41)?

Author Response

The article is valuable. However, the Authors make mistakes that should be corrected. 

Response: Thanks for your advice. The mistakes were carefully checked and revised. You can refer it in the revised manuscript, highlighted in blue.

1 Figure 3 a - unclear description

Response: Thanks for your advice. Figure 3a showed that AGS size distribution before the agnail aeration device employed in batch experiment. The proportions of AGS size on 4-5mm, 3-4 mm, 2-3 mm were 15±0.5%, 25±0.5%, 55±0.5%, decreased to 2±0.5%, 5±0.5% and 41±0.5%, respectively (Figure 3a). But the proportions of AGS size on 1-2mm and 0.2-1 mm and 2-3 mm both increase from 10±0.5% and 0% to 44±0.5% and 8±0.5% after 4 days’ batch experiment (Figure 3b). This means that the agnail device had outstanding restriction on AGS size.

2 Figure 5 - vertical axis SVI (g/mL)? Line 256 Figure 8 b - d? Line 275 - 277 Figure 7? Line 286 (Figure 41)?

Response: Thanks for your questions. The unit of vertical axis SVI is mL/g in Figure 5. The figures of line 256, 275-277 and 286 were figure 7b-d, figure 7d and figure 4k-l, respectively. These mistakes were revised in the revised manuscript, highlighted in blue. You can refer it.

Reviewer 2 Report

Dear authors, I congratulate you for the research done, I recommend minor corrections.

In line 78, you specified figure 1c, but c does not appear in the figure, I recommend you to put it in figure c as well.

From line 345 to 436, I recommend that you write all the sources according to the journal's instructions.

In line 207, you put the square brackets but did not pass the reference.

1. Author 1, A.B.; Author 2, C.D. Title of the article. Abbreviated Journal Name Year, Volume, page range.

2. Author 1, A.; Author 2, B. Title of the chapter. In Book Title, 2nd ed.; Editor 1, A., Editor 2, B., Eds.; Publisher: Publisher Location, Country, 2007; Volume 3, pp. 154–196.

Author Response

1 In line 78, you specified figure 1c, but c does not appear in the figure, I recommend you to put it in figure c as well.

Response: Thanks for your advice. The comb made of plastic (Figure 1a) is usually done one’s hair, and the long hair was dropping during dress up process. Then it is remolded by manual work (Figure 1b), including modification of both ends with two holes of 9 mm for the aeration pipe passing through. finally, the agnail aeration device was successfully obtained (Figure 1c). You can refer it in the revised manuscript, highlighted in blue (Page 4, line 97-101).

Figure 1. Schematic diagram of agnail aeration device by manufacture modification.

2 From line 345 to 436, I recommend that you write all the sources according to the journal's instructions.

Response: Thanks for your advice. All the resources were checked in the manuscript, which was revised with blue colour.

3 In line 207, you put the square brackets but did not pass the reference.

Response: Thanks for your advice. This square bracket was [5, 43] in the manuscript. You can refer it in the revised manuscript, highlighted in blue.

Reviewer 3 Report

The text is interesting from both the scientific and the factual aspects.  A few moderate errors are in the figures. I recommend publishing after correcting these errors.

Detailed notes:

Line 38-34: Sentence „ In the past decades, aerobic granular sludge (AGS) has been extensively studied and 38 used from lab to full-scale bioreactors as a novel biotechnology [1-4]. This is because of 39 AGS advantages on excellent settleability, dense structure, high biomass content and re-40 tention, and resource recovery [5-7]” is very similar to sentence in Liu, J., Yin, S., Xu, D., Piché-Choquette, S., Ji, B., Zhou, X., & Li, J. (2022). Fast Granulation by Combining External Sludge Conditioning with FeCl3 Addition and Reintroducing into an SBR. Polymers, 14(17), 3688.

Please add a schematic of the SBR with an indication of how the aeration device was installed.

Why are low concentrations of N-NH4+ used in the artificial wastewater? Are COD and N-NH4+ concentrations typical for domestic sewage? Due to the tendency to save water, the concentration of N-NH4 in urban wastewater increases up to 100 mg/L.

Figure 1a is unnecessary.

Line 16, 129, 131 – wrong font size.

Figure 3 should be after the results.

Figure 6a – what does the “star” label on the chart mean.

Figure 7a – no chart results for influent COD and N-NH4+.

Where is Figure 8b-d?

Line 276-277 – should be Figure 8.  Figures should be after mentioning them in the text.

Author Response

1 Line 38-34: Sentence „ In the past decades, aerobic granular sludge (AGS) has been extensively studied and 38 used from lab to full-scale bioreactors as a novel biotechnology [1-4]. This is because of 39 AGS advantages on excellent settleability, dense structure, high biomass content and re-40 tention, and resource recovery [5-7]” is very similar to sentence in Liu, J., Yin, S., Xu, D., Piché-Choquette, S., Ji, B., Zhou, X., & Li, J. (2022). Fast Granulation by Combining External Sludge Conditioning with FeCl3 Addition and Reintroducing into an SBR. Polymers14(17), 3688.

Response: Thanks for your advice. This part was rewritten as in the following: As a novel biotechnology, aerobic granular sludge (AGS) has been widely researched in the bioreactors for the decades [1-4]. This is because of its advantages on compact structure, excellent settleability, resource recovery and low energy usage [5-7]. And this part was revised in the manuscript, which was highlighted in blue. You can refer it.

2 Please add a schematic of the SBR with an indication of how the aeration device was installed.

Response: Thanks for your advice. As we know, the comb made of plastic (Figure 1a) is usually done one’s hair, and the long hair was dropping during dress up process. Then it is remolded by manual work (Figure 1b), including modification of both ends with two holes of 9 mm for the aeration pipe (8 mm) passing through. Finally, the agnail aeration device was successfully obtained (Figure 1c). And this device put into the SBR (H×D =100 cm× 10 cm) from its top when it was used in this work. Obviously, the above process was easily done in this experiment. This part was revised in manuscript, which was highlighted with blue. You can refer it.

Figure 1. Schematic diagram of agnail aeration device by manufacture modification.

3 Why are low concentrations of N-NH4+ used in the artificial wastewater? Are COD and N-NH4+ concentrations typical for domestic sewage? Due to the tendency to save water, the concentration of N-NH4+in urban wastewater increases up to 100 mg/L.

Response: Thanks for your questions. In this work, the artificial wastewater had an influent chemical oxygen demand (COD), nitrogen from ammonium (NH4+-N) and total phosphorus (TP) maintained at 500-700 mg·L-1, 25-35 mg/L and 10-15 mg·L-1, respectively. As we know, the domestic sewage COD and NH4+-N are 200-400 mg/L and 20-30 mg/L, which is approached to the artificial wastewater. Meanwhile, the artificial wastewater is usually used in lab-scale SBR for AGS study. If a few fresh waters discharged into domestic wastewater, NH4+-N concertation will increase up to 100 mg/L, even above 200 mg/L. Our earlier works found this wastewater (Liu et al., 2019, 2023).

References

Liu, J.; Li, J.; Xie, K., Sellamuthu, B. Role of adding dried sludge micropowder in aerobic granular sludge reactor with extended filamentous bacteria. Bioresour. Technol. R. 2019, 5, 51-58.

Liu, J.; Xu, D.; He, W.Q.; He, Q.L.; Chu, W.H.; Li, S.B.; Li, J. Results of adding

sludge micropowder for microbial structure and partial nitrification and denitrification in a filamentous AGS-SBR using high-ammonia wastewater. Water 2023, 15, 508.

5 Figure 1a is unnecessary.

Response: Thanks for your advice. In this work, figure showed a comb made of plastic, which used on done one’s hair. Then it is remolded by manual work (Figure 1b), including modification of both ends with two holes of 9 mm for the aeration pipe passing through. finally, the agnail aeration device was successfully obtained (Figure 1c). From this view, figure 1a is the indispensable and retained in this work.

5 Line 16, 129, 131 – wrong font size.

Response: Thanks for your advice. The font size of line 16, 129, 131 were checked. These mistakes were revised in the revised manuscript, highlighted in blue. You can refer it.

6 Figure 3 should be after the results.

Response: Thanks for your advice. In this work, figure 3 showed that results of batch experiment, which followed in the part of “3 Results and discussion”. The part of “3.1 Effects of batch experiment” was the detailed information, you can refer it. In brief, figure 3 after the results in the manuscript.

7 Figure 6a – what does the “star” label on the chart mean.

Response: Thanks for your question. Due to AGS mean size in both SBRs has no difference before day 40, which leads to the data overlapped and showed with “star” in figure 6a. Now, we revised the figure, you can refer it.

Figure 6a. Changes of sludge characteristics on velocity & size.

8 Figure 7a – no chart results for influent COD and N-NH4+.

Response: Thanks for your questions. The artificial wastewater had an influent chemical oxygen demand (COD), nitrogen from ammonium (NH4+-N) and total phosphorus (TP) maintained at 500-700 mg·L-1, 25-35 mg/L and 10-15 mg·L-1, respectively. More data led to the curve overlapping, which presented in figure 7a in terms no chart results of COD and NH4+-N. Now, we revised the figure 7a (in the following) for easy observation the chart in the manuscript.

Figure 7a. Performance in the whole experiment.

9 Where is Figure 8b-d? Line 276-277 – should be Figure 8.  Figures should be after mentioning them in the text.

Response: Thanks for your questions. The figures of line 256 and 276-277 were figure 7b-d and figure 8. These mistakes were revised in the revised manuscript, highlighted in blue. You can refer it.
